# Correlation between the static and dynamic responses of organic single-crystal field-effect transistors

Taiki Sawada[1], Akifumi Yamamura[1], Mari Sasaki[1], Kayo Takahira[1], Toshihiro Okamoto [1,2,3], Shun Watanabe[1,2 ✉] & Jun Takeya [1,4 ✉]

Transistors, the most important logic elements, are maintained under dynamic influence during circuit operations. Practically, circuit design protocols and frequency responsibility should stem from a perfect agreement between the static and dynamic properties. However, despite remarkable improvements in mobility for organic semiconductors, the correlation between the device performances achieved under static and dynamic circumstances is controversial. Particularly in the case of organic semiconductors, it remains unclear whether parasitic elements that relate to their unique molecular aggregates may violate the radio-frequency circuit model. Thus, we herein report the manufacture of micrometre-scale transistor arrays composed of solution-processed organic semiconductors, which achieve near very high-frequency band operations. Systematic investigations into the device geometrical factors revealed that the radiofrequency circuit model established on a solid-state continuous medium is extendable to organic single-crystal field-effect transistors. The validity of this radiofrequency circuit model allows a reliable prediction of the performances of organic radiofrequency devices.

[1] Material Innovation Research Center (MIRC) and Department of Advanced Material Science, Graduate School of Frontier Science, The University of Tokyo, 5-1-5 Kashiwanoha, Kashiwa, Chiba 277-8561, Japan. [2] AIST-Utokyo Operando-Measurement Technology Open Innovation Laboratory (OPERANDO-OIL), National Institute of Advanced Industrial Science and Technology (AIST), 5-1-5 Kashiwanoha, Kashiwa, Chiba 277-8561, Japan. [3] JST, PRESTO, 4-1-8 Honcho, Kawaguchi, Saitama 332-0012, Japan. [4] International Centre for Materials Nanoarchitectonics (WPI-MANA), National Institute for Materials Science (NIMS), 1-1 Namiki, Tsukuba, Ibaraki 205-0044, Japan. ✉email: swatanabe@edu.k.u-tokyo.ac.jp; takeya@k.u-tokyo.ac.jp

Metal-oxide-semiconductor field-effect transistors (MOSFETs) typically based on silicon have revolutionized modern electronics and informatics, where effective scaling and miniaturization has allowed the production of high-density integrated circuits, such as microprocessors and memory devices[1,2]. The development of high-performance MOSFETs requires the simultaneous optimization of various performance parameters, including not only semiconductor parameters, such as the mobility and doping concentrations, but also the geometrical parameters of layered devices, such as the channel length and the oxide thickness[3,4]. These parameters can play a decisive role in determining the electric performance of an individual MOSFET. One of the virtues established in modern MOSFETs manufacture is that the dynamic response in digital and analog circuits can be predictable with the static performance of a single MOSFET, i.e., the device parameters determined under DC operating conditions can be applied to the transient analysis of integrated circuits, which allows an elaborate circuit design of high-density integrated circuits, analog circuits, and functional power devices[3-6]. Compact models for semiconductor devices, supported by simulation program with integrated circuit emphasis (SPICE), are a representative example[7,8]. In the context of their establishment in modern silicon electronics, these sophisticated systems ensure the appropriate circuit design protocols, operational stabilities, and frequency responsibilities of the integrated digital and analog circuits, which should stem from a perfect agreement between the static and dynamic electric properties of a single transistor.

Organic semiconductors (OSCs) are lightweight, flexible, and exhibit a low-temperature processability, which can lead to the production of flexible, functional electronic devices[9,10]. In particular, their unique solution-processability at relatively low temperatures has opened opportunities in the manufacture of printable electronics[11,12]. Following various recent developments in printing technologies and in materials science, the mass production of highly integrated OSC devices is expected to lead to challenges in terms of the internet of things (IoT) challenges[13-15]. Recently, various groups have demonstrated the wafer-scale fabrication of ultra-thin single-crystal OSCs via a one-shot solution process[16-27]. The resulting excellent electronic properties, including a field-effect mobility up to $10 \, \text{cm}^2 \, \text{V}^{-1} \, \text{s}^{-1}$ originating from coherent band-like transport[20,27,28], in conjunction with the miniaturization of organic field-effect transistor (OFET) devices, allows high-speed switching operations at a few tens of a MHz[20,29,30]. Although the electronic properties of organic devices have improved considerably, in-depth understandings regarding a direct correlation between the static and dynamic responses of OFETs remain limited[29,31-35]. In general, the dynamic performance of a transistor is evaluated by the current gain cutoff frequency ($f_T$), which is defined as the characteristic frequency at which the current gain is unity, and is given by $f_T = g_m/2\pi C_G$, where $g_m$ and $C_G$ are the transconductance ($= \partial I_D/\partial V_G$, $I_D$: drain current, $V_G$: gate voltage), and the total gate capacitance, respectively[1-4]. In reality, $C_G$ comprises the gate capacitance and the parasitic capacitance ($C_p$), whereby the latter is caused by an extrinsic effect for which additional currents flow at the gate-overlapped area, but must be taken into account particularly for a transistor with a short channel length ($L$) and contact length ($L_C$)[1-4,20,29-35]. Here, $2L_C$ is the characteristic distance by which the gate electrode overlaps the periphery of the source and drain electrodes. In the saturation region, $f_T$ can be rewritten as[20,29-35]:

$$f_T = \frac{\mu_{\text{eff}}(V_G - V_{\text{th}})}{2\pi L(\frac{2}{3}L + 2L_C)}, \quad (1)$$

where $V_{\text{th}}$ is threshold voltage. According to Eq. (1), $f_T$ can be improved by a large effective mobility ($\mu_{\text{eff}}$), short $L$, and short $L_C$. The scaling and miniaturization of devices can lead to an effective reduction in both $L$ and $L_C$, but in contrast, this is always in a trade-off relationship with $\mu_{\text{eff}}$, since the effects of contact resistance ($R_C$) are no longer negligible in the short channel devices, thereby resulting in a reduction in $\mu_{\text{eff}}$. Although Eq. (1) gives a practical guide to the correlate dynamic response with the static performance, i.e., $f_T$ can be predicted from $\mu_{\text{eff}}$, $L$, and $L_C$, the validity of the radiofrequency circuit model represented by Eq. (1), which has been established for condensed matter semiconductor devices[1-3] remains unclear. In particular, for carbon nanotube transistors, the discrepancy between the static and dynamic responses has often been reported since an additional contribution of the parasitic capacitance between the nanotube-gate electrode of an individual nanotube is known to reduce the theoretical $f_T$[36,37]. It is quantitatively understood that sparse-density nanotubes that are often found in solution-processed thin films behave as the parasitic capacitance, for which even a slight spacing between nanotubes of 100 nm causes $f_T$ to half[36,37]. This motivates the investigation into whether parasitic elements in OFETs related to their unique molecular aggregates may violate the standard radiofrequency circuit model.

In this work, we focus on the manufacture of solution-processed organic single-crystal transistors, and subsequent characterization of the dynamic responses at near very high-frequency band (above 30 MHz). Our original damage-free lithography technique will be expected to allow a fine patterning of electrodes directly on the surfaces of OSCs, with the ultimate aim of producing effective miniaturization of OFETs with a sub-micrometer spatial resolution. Systematic changes in $L$, and $L_C$ in conjunction with full two-port scattering $S$-parameter measurements will be expected to reveal, whether the standard radiofrequency circuit model established on a solid-state continuous medium can be extended to solution-processed OFETs. In addition, we discuss the importance of the transfer length ($L_T$), which is a characteristic length that determines the contact resistance, and by which the static and dynamic performances can be correlated concomitantly. It is expected that the radiofrequency circuit model presented in this work will allow not only reliable predictions of the operational performances of organic radiofrequency devices, but also the realization of practical, high-speed, organic integrated circuits, radiofrequency rectifiers, and low-noise amplifiers.

## Results

**DC characteristics and evaluation of contact resistance.** To assess the static transistor properties, organic single-crystal transistors with a lithographically-defined device geometry were fabricated, and characterized under DC conditions to allow for an accurate determination of the mobility and contact resistance. Our benchmarked OSC 3,11-dinonyldinaphtho[2,3-$d$:2′,3′-$d'$]benzo[1,2-$b$:4,5-$b'$]dithiophene(C$_9$-DNBDT-NW)[20,27,38] (Fig. 1a) was deposited via a continuous edge casting technique[17,20,27], a meniscus-guided coating method[39], to form a single crystalline bilayer film (for details, see the "Methods" section). Top contact, bottom gate OFETs containing electrodes bearing a submicrometre-scale spatial resolution were fabricated lithographically using a fluorinated photoresist and developer (Fig. 1b, c). A similar device structure was reported in our previous studies[20,30], with the exception that the thickness of the gate dielectric layer (AlO$_x$) was reduced to 60 nm to increase the accumulated carrier density at the channel (Fig. 1b, c). We also note that the dual-channel OFET geometry, for which two active transistor channels sharing a common gate and drain electrodes are formed in parallel (Fig. 1d, e), was employed for the purpose

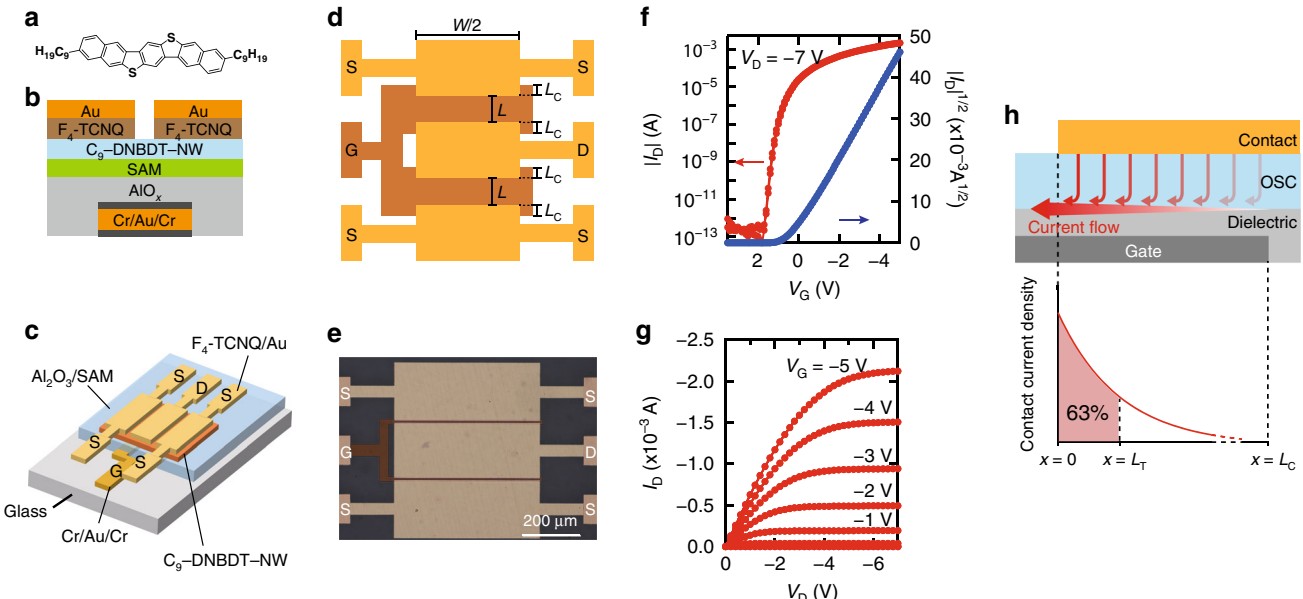

**Fig. 1 Static transistor characteristics obtained under DC conditions. a** Molecular structure of $C_9$-DNBDT-NW. **b** Cross-section and **c** 3D illustration of the top-contact, bottom-gate OFETs used in this study. A top-contact, bottom-gate structure was used. Both static and dynamic characteristics were acquired and compared with single OFETs. **d** Top-view of the OFET with a dual-channel structure. Here, $L$, $L_C$, and $W$ are the channel length, contact length, and channel width. S source, D drain, G gate electrodes. **e** Top-view microscopy image of the present OFET with $L = 6$ μm, $L_C = 5$ μm, and $W = 1000$ μm. Scale bar = 200 μm. **f** Static transfer characteristics ($I_D$ vs. $V_G$) in the linear region with $V_D = -7$ V. **g** Output characteristic ($I_D$ vs. $V_D$). $I_D$: drain current, $V_G$: gate voltage, $V_D$: drain voltage. **h** Schematic diagram illustrating the current crowding. $L_T$ is defined as the characteristic length over which 63% of the charge carrier is injected at the contact and semiconductor interface.

of the full two-port $S$-parameter measurements (typical coaxial ground-signal-ground high-frequency probes were used). A total of 28 OFETs with different $L$ and $L_C$ values were fabricated within a monodomain single crystal, and characterization was carried out to correlate the DC and AC electric performances, while maintaining the other geometrical parameters as constants: channel width ($W$) 1000 μm, OSC thickness 8 nm (bilayer thickness), and dielectric gate thickness 60 nm. It should be noted here that our bilayer single crystalline film covers an area greater than 5 cm × 1 cm, and so all 28 OFETs share an exactly identical single crystalline domain, which allows an unambiguous comparison between the static and dynamic properties. The static device performances were acquired using a semiconductor parameter analyser with controlled applications of the $V_G$ and drain voltages ($V_D$).

Figure 1f, g show the static transfer and output characteristics obtained for a typical OFET $L = 6$ μm and $L_C = 5$ μm under DC conditions, whereby the transfer curve in the saturation region was recorded with the application of a constant drain voltage $V_D = -7$ V. The present OFET with a relatively short $L$ exhibits a textbook-like switching characteristic with negligible hysteresis, a high on-off ratio (>$10^{10}$), and a clear current saturation behavior, with the exception of $V_{th}$. The shift of $V_{th}$ to a transistor ON state (normally-on state) has often been reported in OFETs with short channels, which is presumably because of an unintentional chemical doping at the channel during lithography processes[16]. Although this can be problematic in terms of the device stability in an actual integrated circuit, it should be satisfactory for the characterization of dynamic responses as long as the shift of $V_{th}$ is taken into account during analysis. Next, we discuss the importance of $L_T$, which can project the effect of contact resistance to a length scale. $L_T$ is defined as the characteristic length over which 63% of the charge carrier is injected at the contact and semiconductor interface (Fig. 1h).

From the transfer characteristics, the effective filed-effect mobilities $\mu_{eff}$ were determined to be 8.6 $cm^2 V^{-1} s^{-1}$ in the saturation regime. When $L$ is reduced, the contribution of the contact resistance relative to the total device resistance increases, and so the value of $\mu_{eff}$ in the short channel OFETs does not generally agree with the intrinsic mobility $\mu_{int}$. Here, we investigated the contact resistance effects by means of the transmission line method (TLM), where OFETs bearing different channel lengths $L$ ranging from 2 to 50 μm were fabricated (Fig. 2a, b). Importantly, an active OSC layer for these OFETs consists of a truly monodomain single crystal with a bilayer thickness; therefore, neither the inhomogeneous distribution of $\mu_{int}$, nor spurious effects such as trapping at grain boundaries can contaminate analysis. Figure 2b, c show typical transfer characteristics with various values of $L$ in the linear region ($L_C = 5$ μm, 3 μm, respectively), which confirms that neither hysteresis nor kinks exist, even in the present OFET, which has an $L$ value of only a few μm. Since the monodomain single crystal was used, the values of $V_{th}$ are essentially independent of $L$, except for the case where $L = 1.5$ μm. Hereafter, the shift of $V_{th}$ was taken into account, i.e., the static and dynamic parameters were compared with standardizing $V_G - V_{th}$.

Figure 2d–g show the width-normalized total resistance ($R_{total} \cdot W$) as a function of $L$ with various $V_G - V_{th}$ ($L_C = 5$ μm, and 3 μm, respectively). Since the present bilayer single crystalline film covers a length of more than a few centimetres, the contact resistance can be extrapolated with relatively high accuracy, which is evidenced by a high square of the regression coefficient of 0.99 over the $V_G - V_{th}$. The width-normalized contact resistance ($R_C \cdot W$) was determined from the y-intercept of $R_{total} \cdot W$ vs. $L$ at a given value of $V_G - V_{th}$, as shown in Fig. 2f, g. It was found that $R_C \cdot W$ depends significantly on $V_G - V_{th}$ (Fig. 2h), which is likely to be caused by the trap states in the access region[20,30], whereas $V_G$ invariant $R_C \cdot W$ has been

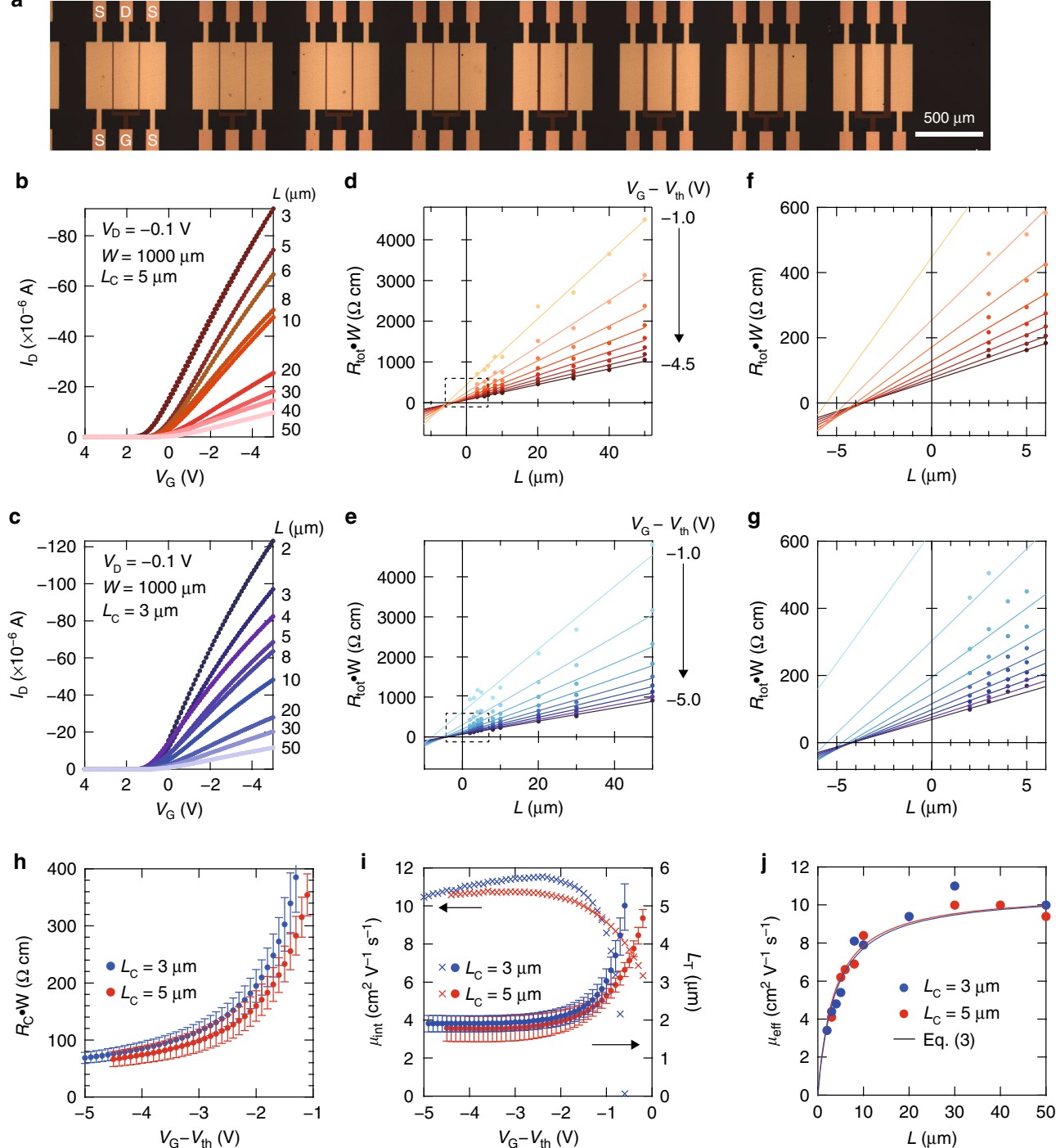

**Fig. 2 Evaluation of the contact resistance and transfer length. a** Top-view photograph of the OFET used for the transfer line method (TLM). Scale bar = 500 μm. Static transfer characteristics in the linear region with different channel lengths $L$ for **b** $L_C = 5$ μm and for **c** $L_C = 3$ μm. **d, e** Corresponding TLM plots for the present OFETs at various values of $V_G - V_{th}$. **f, g** A magnified view at the intercept in **d, e**. Solid lines denote the linear fitting. The width-normalized contact resistance ($R_C \cdot W$) and transfer length ($L_T$) were extracted from the y-intercept and the x-intercept of the linear fittings for the data. **h** Dependence of $R_C \cdot W$ on $V_G - V_{th}$ with various contact lengths ($L_C$). **i** Dependences of the intrinsic mobility ($\mu_{int}$) and $L_T$ on $V_G - V_{th}$ with various values of $L_C$. **j** Dependence of the effective mobility ($\mu_{eff}$) on $L$ with various values of $L_C$. Black curves denote the fitting results based on Eq. (3). The error bars for $R_C \cdot W$ and $L_T$ were determined from uncertainties in the fitting and represent one standard deviation.

previously reported in an ideal Ohmic contact[20,30]. This discrepancy can be an indicative of the imperfection of dopant insertion. Indeed, $R_C \cdot W$ was estimated to be 60–70 Ωcm, which is slightly larger than that obtained for the similar top-contact, bottom-gate device of C$_9$-DNBDT-NW[20,30]. This is presumably because of the poor quality of 2,3,5,6-tetrafluoro-7,7,8,8-

tetracyanoquinodimethane (F4TCNQ) deposition found particularly in the present OFET[20,30]. We also noticed that small molecule dopants such as F4TCNQ require extra care when inserted into the contact. More specifically, the thickness of the dopant layer should be optimized such that the dopant layer itself does not act as a bulk resistance[30]. Although there will be room

for further improvement in the contact resistance obtained herein, this does not cause a fatal error in our analysis as long as $R_C \cdot W$ is assessed accurately. To address the effects of $L_C$ on the contact resistance, an additional key geometrical parameter, namely $L_T$, is introduced[29,31–35,40,41]. $L_T$ is the characteristic length that determines the carrier injection at the contact and semiconductor interface. In principle, carrier injection at a linear regime with a staggered geometry undergoes current crowding[29,31–35,40,41] at an effective injection area $L_T \cdot W$. $L_T$ relates the contact resistivity to the channel sheet resistivity:

$$\frac{R_C \cdot W}{R_{sheet}} = 2L_T \coth\left(\frac{L_C}{L_T}\right), \quad (2)$$

where $R_{sheet}$ is the sheet resistance of the semiconductor layer. Intuitively, a larger part of the contact area can be involved in carrier injection as $R_{sheet}$ is reduced, i.e., the semiconductor channel becomes more conductive. Note that when $L_C$ is assumed to be larger than $L_T$ by a factor of 2 ($L_C > 2L_T$), $\coth\left(\frac{L_C}{L_T}\right)$ in Equation (2) can be approximated to unity, and it may be controversial whether the current crowding model is applicable when $L_C$ is significantly smaller than $L_T$[42,43]. Analytically, $L_T$ can be evaluated from the $x$-intercept of the TLM plots; the value of $L$ that gives $R_{total} \cdot W = 0$ is equal to $-2L_T \coth\left(\frac{L_C}{L_T}\right)$ (Fig. 2f, g). Although previous literature has often highlighted the importance of $L_T$, we hereafter pay more attention to how $L_T$ could play a crucial role in the static and dynamic transistor properties.

Figure 2i shows two essential static parameters, namely the intrinsic mobility ($\mu_{int}$) and the transfer length ($L_T$) as a function of the gate voltage ($V_G - V_{th}$), where the $V_G$ dependence of $L_T$ was determined separately for $L_C = 3$ and 5 µm from the TLM plot shown in Fig. 2d–g. Both $\mu_{int}$ and $L_T$ show similar trends, whereby both exhibit a clear plateau when greater numbers of charge carriers accumulate at the semiconductor channel, i.e., when $|V_G - V_{th}|$ becomes large. This observation suggests that carrier transport at the semiconductor channel approaches the value of $\mu_{int}$, which is free from the effects of contact resistance, thereby resulting in the saturation of $L_T$ (Fig. 2i). It should be noted here that $V_G$-invariant static parameters give confidence that $\mu_{int}$ and $L_T$ can be assumed to be constant during analysis of the dynamic response. The values of $\mu_{int}$ and $L_T$ were determined to be 10.7 cm$^2$ V$^{-1}$ s$^{-1}$ and 1.8 µm, respectively, and most importantly, they were found to be independent of $L_C$. The effective mobility ($\mu_{eff}$) under the influence of the contact resistance can be expressed as follows (for details see the "Methods" section):

$$\mu_{eff} = \mu_{int} \frac{L}{L + 2L_T \coth\left(\frac{L_C}{L_T}\right)} \quad (3)$$

With the determined values of $\mu_{int} = 10.7$ cm$^2$ V$^{-1}$ s$^{-1}$, $L_T = 1.8$ µm, and $L_C = 3$ and 5 µm, the experimentally obtained $\mu_{eff}$ can be reproduced (see solid curves in Fig. 2g). It should be noted that the current crowding model is likely to be violated when OFETs have a short contact length. The present short contact device with $L_C = 1$ µm is therefore likely to be in this violated regime because the obtained $L_T = 1.8$ µm is notably larger than $L_C = 1$ µm[42,43]. Therefore, $L_T$ was not assessed quantitatively for short contact OFETs, but instead we assumed that the experimentally determined $\mu_{int} = 10.7$ cm$^2$ V$^{-1}$ s$^{-1}$ and $L_T = 1.8$ µm can be extrapolated to short contact OFETs. The validity of this assumption will be discussed later. Overall, the obtained results verify that the static transistor properties, even for OFETs with relatively short $L$ values, can be explained based on the conventional current crowding model. More importantly, only two parameters, namely $\mu_{int}$ and $L_T$, which had not previously

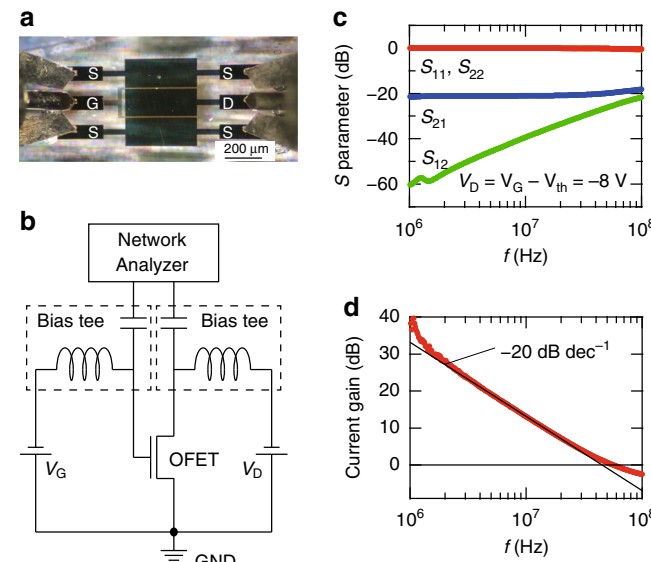

**Fig. 3 Dynamic response evaluated by S-parameter measurements.**
**a** Top-view microscopy image of the dual-channel OFET used for the full two-port scattering S-parameter measurements. **b** Schematic diagram of the full two-port scattering S-parameter measurements. **c** Dependence of the full S-parameters on frequency ($f$) for the OFET with $L = 1.5$ µm and $L_C = 1$ µm. The current gain as a function of $f$ was evaluated from Eq. (5), as described in the "Methods" section. **d** A typical example of the current gain as a function of $f$. $f_T$ is defined as the frequency at which the current gain is zero (20log $|h_{21}| = 0$, equivalently $|h_{21}| = 1$).

been accurately evaluated in short-channel OFETs, in conjunction with priori-fixed $L$ and $L_C$ values, enabled us to describe the static transistor properties, and these parameters could potentially have a close relationship with the dynamic response.

**AC characteristics and evaluation of the cut-off frequency.** The dynamic performance of the transistor was then evaluated to obtain the current gain $f_T$. To probe the high-frequency response of the present OFETs, on-chip high-frequency measurements were performed using a vector network analyser in conjunction with a coaxial ground-signal-ground probe in the range of 1–100 MHz. A standard Open-Load-Short-Thru calibration was employed to de-embed the extrinsic signals from the parasitic capacitance and series resistance associated with the pads and cables. Bias-Tee was used to combine the DC and RF signals (Fig. 3b), which ensures that the OFETs are operating in the saturation region. The de-embedded, full scattering S-parameters comprise a complete set of coefficients of the intrinsic input and the output electrical signals of the OFETs. Figure 3c shows a plot of the four S-parameters as a function of frequency ($f$), by which the short circuit small-signal current gain 20log $|h_{21}|$ (in units of dB) was evaluated (see the "Methods" section). As shown in Fig. 3d, the current gain decreases with increasing $f$, following $-20$ dB dec$^{-1}$ (represented as a black line in Fig. 3d), which is consistent with the conventional model expected for FETs; the observation of a decay slope of $-20$ dB dec$^{-1}$, equivalent to $f^{-1}$ dependence, is a consequence that the gate impedance given as $j\omega C_G$ decreases with increasing frequency. Here, $\omega = 2\pi f$ and $C_G$ is the total gate capacitance. Thus, this gives us confidence that the present full S-parameter measurements are valid, and can be used to evaluate $f_T$. We note that $f_T$ is defined as the frequency at which the current gain is zero (20log $|h_{21}| = 0$, and so equivalent to $|h_{21}| = 1$). According to Eq. (1), a combination of the large $\mu_{eff}$, short $L$, and short $L_C$ gives a large value of $f_T$. Indeed, a large $f_T$ of

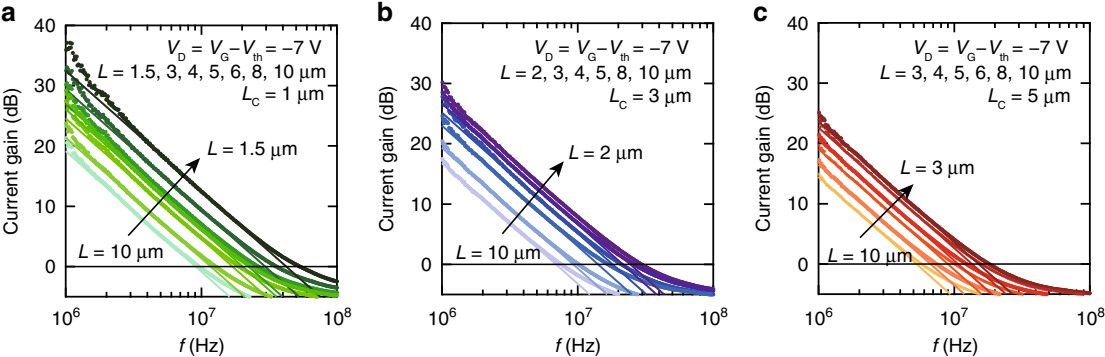

**Fig. 4 Dependence of the cut-off frequency with respect to OFET geometrical factors.** Plots of current gain as a function of frequency with different channel lengths ($L$) and contact lengths ($L_C$). **a** $L_C = 1$ μm, **b** $L_C = 3$ μm, **c** $L_C = 5$ μm.

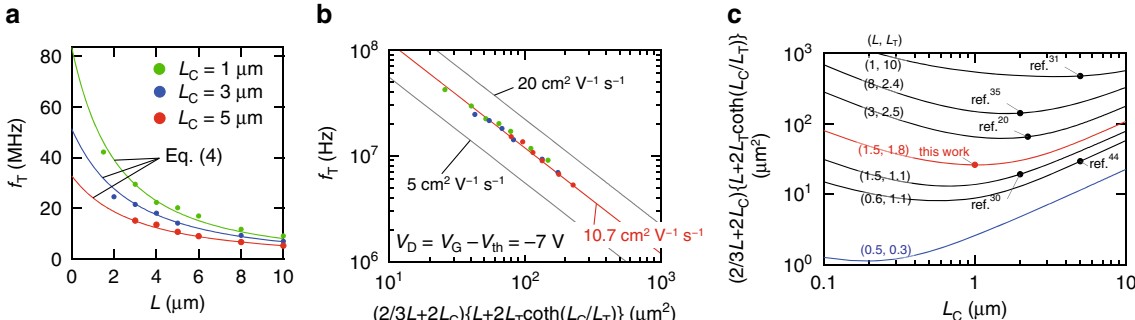

**Fig. 5 Correlation of the static and dynamic electrical performances of the OFETs. a** Dependence of $f_T$ on $L$ with various values of $L_C$. Solid curves represent theoretical $f_T$ values derived from Eq. (4). **b** Dependence of $f_T$ on the OFET areal factor $(\frac{2}{3}L + 2L_C) \cdot \left\{L + 2L_T \coth\left(\frac{L_C}{L_T}\right)\right\}$. The red line represents data with $\mu_{int} = 10.7$ cm$^2$ V$^{-1}$ s$^{-1}$. **c** Dependence of the areal factor on $L_C$ with various combinations of $L$ and $L_T$. Values in parentheses represent $L$ and $L_T$. Solid curves indicate the theoretical areal factors that are uniquely defined when three length parameters are given. Red curve: data calculated with the experimentally determined values of $L$ and $L_T$. Black curves: data calculated using $L$ and $L_T$ values estimated from previous studies. Blue curve: data that reproduces the areal factor of 1 μm$^2$. Circles plotted on each curve represent the experimentally obtained areal factors. Areal factors are adapted from references: 472 μm$^2$ ($f_T = 2.2$ MHz) for Ante et al. (Max Planck group)[31], 140 μm$^2$ (no $f_T$ data) for Borchert et al. (Max Planck group)[35], 63 μm$^2$ ($f_T = 20$ MHz) for Yamamura et al. (this group)[20], 29 μm$^2$ ($f_T = 21$ MHz) for Borchert et al. (Max Planck group)[44], 26 μm$^2$ ($f_T = 45$ MHz) for this work, and 19 μm$^2$ ($f_T = 38$ MHz) for Yamamura et al. (this group)[30].

45 MHz was achieved with the short-channel OFET where $L = 1.5$ μm and $L_C = 1$ μm, which is the largest value reported to date for OFETs and sufficient enough for use as a wireless power supply for the near-field communication (13.56 MHz) RFID tags. Furthermore, the $f_T$ values for 19 transistors with different $L$ and $L_T$ values were evaluated systematically (Fig. 4a–c). Similarly, as shown in Fig. 3, the current gains for all 19 OFETs exhibited $f^{-1}$ dependence, from which $f_T$ was evaluated. It should be noted that $\mu_{int}$ and $L_T$ were initially determined within a monodomain of the single crystalline films, and were found to be independent regardless of device geometry, thereby allowing us to link the static parameters $\mu_{int}$ and $L_T$ to the dynamic parameter $f_T$.

Using the experimentally determined values of $\mu_{int}$ and $L_C$, the expression of $f_T$ can be written as follows (see the "Methods" section):

$$f_T = \frac{\mu_{int}(V_G - V_{th})}{2\pi(\frac{2}{3}L + 2L_C)\left\{L + 2L_T \coth\left(\frac{L_C}{L_T}\right)\right\}},\qquad(4)$$

Importantly, all parameters on the right-hand side of Eq. (4) are experimentally-addressable, static parameters. To validate this, $f_T$ was plotted as a function of $L$ with various $L_C$ values (Fig. 5a), whereby the solid curves represent a theoretical $f_T$ derived from Eq. (4) with $\mu_{int} = 10.7$ cm$^2$ V$^{-1}$ s$^{-1}$ and $L_T = 1.8$ μm, and show an excellent agreement with the experimental $f_T$ value. Similarly, a

universal trace was observed when $f_T$ was displayed as a function of the areal geometrical factor $(\frac{2}{3}L + 2L_C)\left\{L + 2L_T \coth\left(\frac{L_C}{L_T}\right)\right\}$ (in units of μm$^2$) as shown in Fig. 5b, where all experimentally-determined values of $f_T$ can be plotted on the universal line represented by the red line ($\mu_{int} = 10.7$ cm$^2$ V$^{-1}$ s$^{-1}$). A good agreement between the experimental and theoretical $f_T$ values confirms that Eq. (4) is valid, and that the dynamic responses of the OFETs can be predicted using the static device parameters. A slight deviation found particularly in the OFETs with short values of $L$ and $L_C$ may be indicative of an underestimation of $f_T$.

Based on the above results, we could summarize our key experimental findings. More specifically, the two static device parameters, namely the intrinsic mobility ($\mu_{int}$) and the transfer length ($L_T$), can be evaluated accurately using the TLM. The former of these parameters is identical to the mobility purely unique to the material parameter of C$_9$-DNBDT-NW, while the latter is the characteristic length parameter that visualizes how the effects of contact resistance can be dominant, and can be projected on the length scale. Using the priori-designed channel length ($L$) and the contact length ($L_C$), the areal factor $(\frac{2}{3}L + 2L_C) \cdot \left\{L + 2L_T \coth\left(\frac{L_C}{L_T}\right)\right\}$ was introduced for the first time to reproduce the effective mobility ($\mu_{eff}$) under the influence of the contact resistance and the cutoff frequency ($f_T$). Here, the

smallest areal factor achieved in this work was 26 μm², where $L$ = 1.5 μm, $L_C$ = 1 μm, and $L_T$ = 1.8 μm.

Although minimization of the areal factor could be considered the key factor for improving the $f_T$, three length parameters, namely $L$, $L_C$, and $L_T$, clearly interplay to determine the areal factor, which gives a degree of freedom in the geometrical design of OFETs. We would also like to clarify some practical guidelines that have been already highlighted, in addition to some that have not been recognized. Firstly, $L_T$ is a predominant parameter in terms of impacting the whole areal factor, since the term $\coth\left(\frac{L_C}{L_T}\right)$ diverges rapidly as $L_T$ increases. Therefore, a reduction in the contact resistance is necessary. Secondly, a reduction of $L$ without scaling down $L_T$ has a lesser impact on reduction of the areal factor. For example, when $L_T$ is limited to 1.8 μm (the best value obtained herein) only half of the areal factor (14 μm²) is expected to be reached, even when $L$ is 0.01 μm, and this is clearly an ineffective miniaturization despite $L$ being reduced by a factor of 100. In other words, scaling down $L$ and $L_C$ is more effective until these two values become comparable to $L_T$. Thirdly, an optimum $L_C$ can be found once $L_T$ is fixed. As shown in Fig. 5c, the minimum value of the areal factor appears when $L_C$ reaches approximately 70% of $L_T$. Our analysis, by which the areal factor is highlighted for the first time, gives practical guidelines to maximize $f_T$, particularly as the latter two guidelines had not been specifically mentioned in previous studies. Finally, we quantified the key areal factor to achieve higher $f_T$ values in the OFETs. More specifically, a 100 MHz operation requires an areal factor of approximately 10 μm², while 1 GHz requires an areal factor of 1 μm² when $\mu_{int}$ = 10 cm² V⁻¹ s⁻¹, in addition to an input voltage of 5 V. Previously, the smallest reported areal factor was evaluated as approximately 20 μm²[30,44]. In the context of materials science, further improvement of the mobility in conjunction with an effective miniaturization of OFETs is necessary and continues to be an ongoing challenge.

## Discussion

We accurately evaluated both the intrinsic mobility and the transfer length of micrometer-scale OFETs by means of the transmission line method, and successfully correlated these two static parameters with the cut-off frequency determined by de-embedded, full scattering S-parameters. The areal factor, which was introduced for the first time in this study, helped to clarify the one-to-one relationship between the static and dynamic parameters in the electrical performances of OFETs, and showed that the radiofrequency model established in a solid-state, continuous medium is extendable to organic single-crystal transistors. Although the validity of this radiofrequency circuit model is limited only to OFETs with single-crystalline OSCs, it allows a reliable prediction of the performances of organic radiofrequency devices. The areal factor can project electrical parameters, such as the contact resistance, on a length parameter, and visualize the effective miniaturization of OFETs, which is expected to provide a useful design guideline not only in terms of further improvements in the cut-off frequency, but also in the integration of organic digital and analog circuits.

## Method

**Device fabrication**. All devices were fabricated on a pre-cleaned glass substrate. Thermally evaporated Cr (1.5 nm)/Au (20 nm)/Cr (1.5 nm) layers were patterned by conventional photolithography to form gate electrodes, where a standard positive photoresist (TLOR, Tokyo Ohka Kogyo Co., Ltd.) and developer (NMD-3, Tokyo Ohka Kogyo Co., Ltd.) were used. A 60 nm-thick layer of aluminum oxide was deposited as the gate dielectric layer via atomic layer deposition. The surface of the aluminum oxide was then treated with a 2-(phenylhexyl)phosphonic acid self-assembled monolayer (SAM) by immersing the substrate into a 0.2 mM solution for 18 h. Single crystalline bilayer films of C₉-DNBDT-NW were fabricated from a 0.02 wt% 3-chlorothiophene solution with continuous edge casting[20,30]. The substrate

was heated to 81 °C and moved with a shearing rate of 16 μm s⁻¹ while a blade that sustains the meniscus was fixed 95 μm above the substrate. Bilayer single crystalline films were grown selectively by tuning the substrate temperature[45]. Following annealing of the substrate at 100 °C under vacuum to remove any residual solvent, F4TCNQ and Au were subsequently deposited to form the source/drain electrodes, which were patterned by multiple lithographic processes, whereby a negative photoresist (OSCoR4001, Orthogonal inc.) and Au etchant (AURUM S-50790, Kanto Chemical Co. Inc.) were used. Note that F4TCNQ dopants directly above the channel are removed during the Au etching process, so that the active OSC at the channel remains as an intrinsic semiconductor (undoped semiconductor). All photo-exposure processes were carried out using a maskless aligner (MLA150, Heidelberg Instruments) in a clean room environment. The patterning spatial resolution was evaluated as ±300 nm, which was taken into account during analysis.

**Electrical measurements**. All electrical measurements were performed under ambient conditions. The static transistor properties were acquired using a semi-conductor parameter analyser (Keithley 4200-SCS) in conjunction with a manual probe station. The on-chip high frequency measurements employed for evaluation of the cut-off frequency ($f_T$) were conducted using a vector network analyser (Agilent E5061B) combined with a manual high frequency probe station, whereby a coaxial ground-signal-ground high frequency probe (GGB Picoprobe 40A-GSG-200 EDP) was used. Before carrying out any measurements, a standard Short-Open-Load-Thru calibration was completed with a standard calibration substrate (GSG CS-5). Bias-Tee (Mini Circuits, ZFBT-4R2GW) was used to combine the DC and RF signals. A source measurement unit (Keithley 2636) was used as an additional DC voltage supply. The short circuit small-signal current gain was defined as 20log$|h_{21}|$, where $h_{21} = \frac{i_D}{i_G}$ was determined from

$$|h_{21}| = |\frac{-2S_{21}}{(1 - S_{11})(1 + S_{22}) + S_{12}S_{21}}|, \tag{5}$$

where $S_{11}$, $S_{12}$, $S_{21}$, and $S_{22}$ are the de-embedded full scattering S-parameters. In addition, $f_T$ was evaluated by fitting the $f$ dependence of the current gain (Fig. 4d), where $f_T$ was defined as the frequency at 20log$|h_{21}|$ = 0, where $|h_{21}|$ = 1.

**Analysis of $\mu_{eff}$ and $f_T$**. We proposed that the intrinsic mobility ($\mu_{int}$) and the transfer length ($L_T$) are universal parameters that are independent of the device geometry, thereby enabling a direct link between the static and dynamic electrical properties, as outlined in Eqs. (3) and (4). Equation (3) is derived from:

$$\mu_{eff} = \mu_{int}\frac{1}{1 + \frac{R_C \cdot W}{L}C_i\mu_{int}(V_G - V_{th})}, \tag{6}$$

Here, the term $C_i\mu_{int}(V_G - V_{th})$ is equivalent to $(R_{sheet})^{-1}$. Thus, Eq. (6) can be reduced to

$$\mu_{eff} = \mu_{int}\frac{L}{L + \frac{R_C \cdot W}{R_{sheet}}}, \tag{7}$$

Equation (7) is identical to Eq. (3) considering the given expression of $L_T$ in Eq. (2). Combining Eqs. (1) and (3) gives Eq. (4).

## Data availability

The data that support the plots within this paper and other findings of this study are available from the corresponding author (Shun Watanabe; swatanabe@edu.k.u-tokyo.ac.jp) upon reasonable request.

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

## Acknowledgements

S.W. was supported by a Grant-in-Aid for JSPS (Japan Society for the Promotion of Science) Research Fellows. T.O. wishes to thank PRESTO-JST "Molecular Technology and Creation of New Functions" (Grant No. JPMJPR13K5) for financial support. This work was supported by JSPS KAKENHI grant nos. JP26105008, JP17H06123, JP17H06200, and 20H00387.

## Author contributions

T.S. and A.Y. fabricated devices and performed the measurements. M.S. and K.T. assisted with the device fabrications. T.S., A.Y., and S.W. analyzed the data and wrote the manuscript. T.O. synthesized and purified the C$_9$-DNBDT-NW. S.W. and J.T. supervised the work. All the authors discussed the results and reviewed the manuscript.

## Competing interests

The authors declare no competing interests.
