## [Peer Review File · Nature Communications]

Reviewers' Comments:

Reviewer #1:

Remarks to the Author:

The paper presents the fabrication of micrometer-scale OTFT arrays which are capable of high-frequency band operations. The systematic investigation of geometrical factors of the device establishing the relationship between static and dynamic properties. The radiofrequency circuit model are the main novel features claimed in the work. The mathematical model proposed in this study, especially the relationship between the static and dynamic properties of the device, is likely to be discussed in related studies. However, the topic of radiofrequency application of OTFT array is far from recent interest. The radiofrequency circuit model presented by the authors is clearly an improvement over the existing model, but the current version of manuscript is not enough to attract great attention to be published in Nature Communications. A future version of this work might be suitable for publication if several issues are addressed:

1. It would be much impactful if authors could specifically highlight the possibilities of using the proposed model in the electronics field. For example, can authors suggest a specific radiofrequency device suitable for OTFT array and a goal of OTFT performance that should be achieved for realizing this device? If so, how can the presented circuit model can be utilized?
2. Can the same model be applied to inorganic transistors currently used for radiofrequency devices? Can the proposed model be applied for transistors with channels as small as a few nanometers?
3. The performance degradation issues are always existing in utilization of OTFT array, so it is essential to consider the encapsulation. How do authors expect the effect of parasitic capacitance, if encapsulation layer is introduced, to affect the presented model?
4. Authors claimed that the achieved f_T of 45 MHz is the largest value reported to date for OTFTs. Please compare the current achievement with those reported before and comment on whether they are suitable for radiofrequency device applications. Also, additional comparison of the reported f_T with that of inorganic transistors may be beneficial to highlight the author's achievements.
5. As the main variables for the cut-off frequency model, channel length (L) and the contact length (LC) along with the transfer length (LT) are frequently mentioned. Figure 1d gives a clear understanding of LC and L, whereas only a conceptual explanation was provided for LT, making it difficult to understand intuitively. Please provide schematic illustration to help readers understand LT intuitively.

Reviewer #2:

Remarks to the Author:

This manuscript systematically investigates the correlation between the static and dynamic responses of organic single-crystal field-effect transistors. This work successfully correlates the intrinsic mobility and transfer length of OFETs with the cut-off frequency determined by de-embedded, which could help us clarify the one-to-one relationship between the static and dynamic parameters in the electrical performances of OFETs. These interesting results (such as the radiofrequency circuit model of Equation (4)) provide an efficient way to predict the performances of organic radiofrequency devices and a useful design guideline of further improvements in the integration of organic circuits. The manuscript is well organized and written, which could be acceptable for publication after minor revisions. Additional comments are as follows.

- (1) The V_D of transfer characteristics (Figure 1e) is much larger than V_G , which couldn't reflect a

reliable trend of transfer curve and accurate extraction of mobility. Reasonable voltage ranges of V_D and V_G should be applied.

(2) More OFETs based on various OSCs could be demonstrated to reflect the universality of the "radiofrequency circuit model" proposed by this work.

(3) The work investigated the correlation between the static and dynamic responses of a chemically doped single-crystal OFETs. In fact, most OFETs are based on undoped single crystals. Will the result be different in the undoped single crystal?

Reviewer #1

We are grateful for the positive evaluation particularly of our experimental results, and also thank the reviewer for pointing out some technical issues. Let us clarify the radiofrequency model proposed in this work; this model is based on the one employed widely in Si electronics. Most importantly, usage of this well-established radiofrequency model has been controversial for many years in molecular semiconductors such as CNTs, polymers, nanoparticles and so on. In particular, effects of relatively large contact resistance and parasitic capacitance, which are not a crucial factor in inorganic semiconductors, are likely to violate the standard RF model. In fact, discrepancies between theory and experiments are often found in CNT-based transistors (refs. 36,37 in the original manuscript). Here, in this work, we presented and validated a versatile RF model to be able to comprehensively understand dynamic responses in molecular semiconductors. The reviewer's comments are very helpful to gain readability of this work. We appreciate the reviewer #1 for his (her) constructive comments, and carefully considered his/her suggestions, and revised the manuscript accordingly.

(Reviewer's Comment 1)

It would be much impactful if authors could specifically highlight the possibilities of using the proposed model in the electronics field. For example, can authors suggest a specific radiofrequency device suitable for OTFT array and a goal of OTFT performance that should be achieved for realizing this device? If so, how can the presented circuit model can be utilized?

(Our Reply 1)

The reviewer's comment makes a perfect sense that should be clarified more. The RF model proposed in this work can correlate static device properties such as mobility and device geometry to dynamic responses in OTFTs. Clearly, this one-to-one relation between static and dynamic parameters is one of the virtues established in modern Si electronic, *i.e.*, the device parameters determined under DC operating conditions can be applied to the transient analysis of integrated circuits, which allows an elaborate circuit design of high-density integrated circuits and functional power devices. To the best of our knowledge, we obtained the record-high cutoff frequency of 45 MHz for organic FET in this manuscript, which is improved by approximately 20% than the previous record (38 MHz) obtained by our group. This frequency is more than three times faster

than the near-field communication of RFID tags (13.56 MHz). The present study enables the appropriate circuit design protocols, operational stabilities, and frequency responsibilities of the organic integrated circuits.

We have revised the sentences in Introduction and Conclusion parts accordingly.

(Reviewer's Comment 2)

Can the same model be applied to inorganic transistors currently used for radiofrequency devices?

Can the proposed model be applied for transistors with channels as small as a few nanometers?

(Our Reply 2)

Again, the radiofrequency model proposed in this work is based on the one employed widely in Si electronics. Our attempts are to extend this standard model to molecular semiconductors, which has been a central issue for many years. Technically, the proposed model is applicable for transistors with a-few-nanometer channels. However, as has been concluded in the original manuscript, a reduction of channel length without scaling down transfer length has a less impact on improvement of cutoff frequency. Currently, the record-low transfer length is on the order of micrometers for organic semiconductors, hence, the channel length with submicrometer will be the next target. Our model gives intuitive guidelines for an effective miniaturization for OFETs.

(Reviewer's Comment 3)

The performance degradation issues are always existing in utilization of OTFT array, so it is essential to consider the encapsulation. How do authors expect the effect of parasitic capacitance, if encapsulation layer is introduced, to affect the presented model?

(Our Reply 3)

The reviewer has pointed out another important issue. We are also interested in the device lifetime. Since our bench-marked material, C_n-DNBDT-NW, has been synthesized in 2014, we have investigated secular changes in C_n-DNBDT-NW transistors from time to time. Surprisingly, most of them survive with keeping their excellent mobility up to 10 cm²V⁻¹s⁻¹; the lifetime of their transistor operation is estimated to be more than 5 years. We are now collecting a concrete data set for device durability by means of acceleration test. Let us share the important data (private communication). The

attached figure shows the result from acceleration test, where the fabricated C_n -DNBDT-NW FET has been stored at 85 degC and 85 percent relative humidity (85/85 test) without any passivation. It was found that the fabricated FET shows an excellent transistor performance after 143 hours storage at the 85/85 condition. Because we have not had the concrete data set yet, we prefer to keep this result in private communication. Details of device lifetime investigation will appear elsewhere shortly.

The reviewer's comment is correct that an encapsulation layer will be introduced not only for preservation of device performance, but also for device integration. Normally, we employ an organic encapsulation layer, for example parylene, and design this encapsulation layer not to behave as an additional parasitic capacitance, *i.e.*, we ensure that no electrodes face each other via this encapsulating layer. Therefore, we believe that the encapsulation layer has negligible impact on the cutoff frequency.

(Reviewer's Comment 4)

Authors claimed that the achieved f_T of 45 MHz is the largest value reported to date for OTFTs. Please compare the current achievement with those reported before and comment on whether they are suitable for radiofrequency device applications. Also, additional comparison of the reported f_T with that of inorganic transistors may be beneficial to highlight the author's achievements.

(Our Reply 4)

We agree with the reviewer that this point should be clarified. We have added values of cutoff frequency in the caption of Fig.5 (together with areal factors). To the best of our knowledge, we obtained the record-high cutoff frequency of 45 MHz for organic FET in this manuscript, which is improved by approximately 20% than the previous record (38 MHz) obtained by our group.

This frequency is more than three times faster than the near-field communication of RFID tags (13.56 MHz).

We have revised the manuscript accordingly.

In terms of comparison with the cutoff frequency of inorganic transistors, we do not go deep in the details because obviously the cutoff frequency of inorganic transistors is much larger than those of organic transistors. We would like to note that a few tenth of MHz has been obtained for “solution-processed” inorganic semiconductors (IGZO, IZO etc). This kind of comparisons has appeared in several review article (ref. 29 in the original manuscript).

(Reviewer’s Comment 5)

As the main variables for the cut-off frequency model, channel length (L) and the contact length (LC) along with the transfer length (LT) are frequently mentioned. Figure 1d gives a clear understanding of LC and L, whereas only a conceptual explanation was provided for LT, making it difficult to understand intuitively. Please provide schematic illustration to help readers understand LT intuitively.

(Our Reply 5)

We appreciate the reviewer for pointing this out. We have added the figure 1 h and resized the panels in figure 1 accordingly.

Reviewer #2

We are grateful for the positive evaluation particularly of our experimental results and we have carefully considered his/her suggestions and revised the manuscript accordingly.

(Reviewer's Comment 1)

The V_D of transfer characteristics (Figure 1e) is much larger than V_G , which couldn't reflect a reliable trend of transfer curve and accurate extraction of mobility. Reasonable voltage ranges of V_D and V_G should be applied.

(Our Reply 1)

Because the threshold voltage V_{th} is shifted to positive (approximately + 2 V) for the present device, the current condition $V_D = V_G - V_{th} = -7$ V satisfies the requirement for the saturation regime in a transistor. It is evident that a clear saturation behavior is observed at $V_D = -7$ V, and $V_G = -5$ V in output characteristics (Fig.1g). The unintentional V_{th} may attribute to imperfection either of lithography process, or of contact doping. It does not impact on the validity of proposed RF model because the shift of V_{th} is taken account for our analysis.

(Reviewer's Comment 2)

More OFETs based on various OSCs could be demonstrated to reflect the universality of the "radiofrequency circuit model" proposed by this work.

(Our Reply 2)

In this study, we fixed the organic semiconductor material and investigated the geometrical effect on the cutoff frequency. Because the intrinsic mobility needs to be constant over several devices with different channel length and contact length, it should be reasonable to fix the organic

semiconductor material. Recently, the universality of model is further investigated using n-type organic semiconductor, and it is found that the model proposed in this work is applicable to n-type organic single crystal semiconductors (Paper under review by Advanced Materials).

(Reviewer's Comment 3)

The work investigated the correlation between the static and dynamic responses of a chemically doped single-crystal OFETs. In fact, most OFETs are based on undoped single crystals. Will the result be different in the undoped single crystal?

(Our Reply 3)

The reviewer's concern about chemical doping should be clarified. F4TCNQ dopants directly above the transistor channel are removed during the Au etching process. The observation of low off-current in the present OFETs (Figs.1 e and f) supports that the active OSC at the channel remains as an intrinsic (undoped) semiconductor. The present chemical doping with molecular dopants is not strong, and merely shifts the threshold voltage V_{th} . Validity of RF model present in this study does not suffer from the unintentional V_{th} shift because the model is constructed with standardizing $V_G - V_{th}$. We revised the sentence in Experimental section accordingly.

Editorial changes

List of changes (Main changes are highlighted as a blue text)

Main text

- At line 9 on page 3, sentences have been modified.
- At line 22 on page 6, sentences have been added.
- At line 3 on page 10, sentence has been added.
- At line 9 on page 12, sentence has been added.
- At line 1 on page 13, sentences have been added.
- Figure 1 h has been added, and the entire figure 1 has been resized.
- The caption of Fig. 1h has been added.
- the values of cutoff frequency have been added in the caption of Fig.5.

Reviewers' Comments:

Reviewer #1:

Remarks to the Author:

The authors fully addressed the comments from reviewers. This manuscript can be published in Nat. Commun. as it is.

Reviewer #2:

Remarks to the Author:

In the response letter, the authors have responded most of my concern question. While one question the author should further give response and additional supporting data. This work proposed a general radiofrequency circuit model demonstrated only by an OSC which can't reflect the universality of this model. The universality of model should be further investigated using other organic semiconductors and relevant experimental data need to be listed carefully.

[Reply to the Comments by the Reviewers] (Manuscript number: NCOMMS-20-18873A)

=====

Reviewer #1

=====

We thank the reviewer for his/her constructive comments.

Reviewer #2

We thank the reviewer for his/her constructive comments. We have carefully considered his/her suggestions and revised the manuscript accordingly.

(Reviewer's Comment)

In the response letter, the authors have responded most of my concern question. While one question the author should further give response and additional supporting data. This work proposed a general radiofrequency circuit model demonstrated only by an OSC which can't reflect the universality of this model. The universality of model should be further investigated using other organic semiconductors and relevant experimental data need to be listed carefully.

(Our Reply)

We understood the reviewer's concern. Unfortunately, it is not possible at this stage to provide the full data set necessarily to validate the universality of this radiofrequency circuit model. This is because it may take several months or more to acquire large amount of static and dynamic data with different geometrical factors. Alternatively, we analyzed previous data set shown in the ref. 33 (this is the only study showing the full data set with absolute accuracy). Putting these data from previous studies into our proposed model, it actually works well (see the figure in private communication).

Figure S1 Dynamic properties for DNNT-OFETs.

The material used in the ref.33 is a small molecule organic semiconductor (DNNT). Note that thermally-evaporated DNNT forms not a single crystal, but rather a polycrystal. Therefore, we are at least confident that the model proposed in this study does not depend on OSC's crystallinity. We agree

with the reviewer and editor that it is still insufficient to validate the universality of this model. Therefore, according to the editor's suggestion, we decide to tone down the argument regarding universality. The validity of this model is limited only to OFETs with a *single-crystalline* organic semiconductor.

Editorial changes

List of changes (Main changes are highlighted as a blue text)

Main text

- In Abstract, sentences “and is an issue in terms of realizing organic integrated circuit” and “as well as the justification of an effective mobility pursued under DC conditions” were deleted. To clarify the universality, the validity of this model is limited to organic *single-crystal* field-effect transistors.

- At line 7 on page 12, we also limited the validity of this model to organic *single-crystal* field-effect transistors.